# Serum Periostin Level and Genetic Polymorphisms Are Associated with Vertebral Fracture in Chinese Postmenopausal Women

**DOI:** 10.3390/genes13030439

**Published:** 2022-02-27

**Authors:** Yi-Ming Guo, Jian-Hao Cheng, Hao Zhang, Jin-Wei He, Hua Yue, Wei-Wei Hu, Jie-Mei Gu, Yun-Qiu Hu, Wen-Zhen Fu, Chun Wang, Zhen-Lin Zhang

**Affiliations:** Shanghai Clinical Research Center of Bone Disease, Department of Osteoporosis and Bone Disease, Shanghai Jiao Tong University Affiliated Sixth People’s Hospital, 600 Yishan Road, Shanghai 200233, China; guoyiming1@alumni.sjtu.edu.cn (Y.-M.G.); chengjianhao59@gmail.com (J.-H.C.); zhanghao2@medmail.com.cn (H.Z.); he_jinwei@live.com (J.-W.H.); yueyinglonghua@163.com (H.Y.); huweiwei@medmail.com.cn (W.-W.H.); jiemeigu@hotmail.com (J.-M.G.); hu_yunqiu@hotmail.com.cn (Y.-Q.H.); fu_wenzhen@outlook.com (W.-Z.F.); wangchun66@sjtu.edu.cn (C.W.)

**Keywords:** postmenopausal osteoporosis, vertebral fracture, periostin, single nucleotide polymorphism, bone mineral density

## Abstract

Purposes: In order to investigate the association between serum periostin levels and the variation of its encoding gene *POSTN* and the prevalence of vertebral fractures and bone mineral density (BMD) in Chinese postmenopausal women, an association study was performed. Materials and Methods: 385 postmenopausal women were recruited. For participants without a history of vertebral fracture, lateral X-rays of the spine covering the fourth thoracic spine to the fifth lumbar spine were performed to detect any asymptomatic vertebral fractures. Ten tag-single nucleotide polymorphisms (SNP) of *POSTN* were genotyped. Serum periostin levels, biochemical parameters, and BMD were measured individually. Results: rs9603226 was significantly associated with vertebral fractures. Compared to allele G, the minor allele A carriers of rs9603226 had a 1.722-fold higher prevalence of vertebral fracture (*p* = 0.037). rs3923854 was significantly associated with the serum periostin level. G/G genotype of rs3923854 had a higher serum periostin level than C/C and C/G (67.26 ± 19.90 ng/mL vs. 54.57 ± 21.44 ng/mL and 54.34 ± 18.23 ng/mL). Furthermore, there was a negative correlation between the serum level of periostin and BMD at trochanter and total hip. Conclusion: Our study suggested that genetic variation of *POSTN* could be a predicting factor for the risk of vertebral fractures. The serum level of periostin could be a potential biochemical parameter for osteoporosis in Chinese postmenopausal women.

## 1. Introduction

Osteoporosis is a systemic bone disease characterized by low bone mass and the structural destruction of bone. One of the severe osteoporosis consequences is vertebral fractures, which are often undiagnosed due to mild symptoms [1]. Existing vertebral fractures can also lead to a higher risk of non-vertebral fractures in the future [2]. Bone mineral density (BMD), which is assessed by DXA, is often used for the diagnosis of osteoporosis. Although X-ray of the thoracic and lumbar spine is more commonly used in patients with suspected vertebral fractures, recent developments in BMD evaluation have heightened the need for the assessment of pre-existing of vertebral fractures [3]. How to find individuals with high risk of osteoporotic fracture is very important in order to reduce the prevalence of fractures in postmenopausal women. 

Periostin is known for its functional roles in bone formation and fracture healing. Periostin now can be measured in serum and plasma. In the field of bone metabolism, current studies show that serum periostin levels affect the proliferation and differentiation of osteoblasts, as well as other related physiological activities [4,5]. *POSTN* polymorphism is also confirmed to be associated with BMD in animal experiments: when compared with wild mice, knockout *P**ostn* mice had a significantly lower BMD and thinner bone cortex [4,5,6]. In current clinical research, serum periostin level is considered as a potential marker of osteoporotic fracture and higher serum periostin levels tend to have an increasing risk of fractures in the future [7,8]. The clinical studies on association between polymorphism of *POSTN* and BMD and vertebral fractures are quite few. Recent studies show that SNP rs9547970 in *POSTN* has association between BMD and vertebral fractures [9]. The association between bone microstructure and periostin expression has also been revealed [10]. However, the relationship of *POSTN* polymorphism and serum periostin to the risk of vertebral fracture has not been explored in clinical settings [11]. The relationship between the genetic variation of *POSTN*, the serum level of periostin, the risk of vertebral fracture and BMD has not been clarified in Chinese postmenopausal women. 

## 2. Methods

### 2.1. Study Population and Diagnosis

In the study, 385 Han ethnicity postmenopausal women were enrolled from the Department of Osteoporosis and Metabolic Bone Diseases of Shanghai Jiao Tong University Affiliate Sixth People’s Hospital. All participants have been diagnosed with osteoporosis/osteopenia. The study was approved by the Ethics Committee of Shanghai Jiao Tong University Affiliated Sixth People’s Hospital, and each participant signed a written informed consent. Inclusion criteria for the study were: (1) All patients were over 45 years old; (2) the age for natural menopause was older than 40 years; (3) all patients had been in menopause for at least one year. The following patients were excluded: (1) patients with chronic diseases in vital organs: liver cirrhosis, chronic kidney disease (CKD), chronic obstructive pulmonary disease (COPD), chronic digestive diseases (such as Crohn’s disease, peptic ulcers, etc.), tumors, etc.; (2) patients who have taken or have been taking drugs with an impact on bone metabolism (such as glucocorticoids, anticonvulsant drugs such as phenytoin sodium, estrogen, etc.); (3) patients with other metabolic bone diseases (such as osteomalacia, osteosclerosis or other monogenic bone disease); (4) patients with other endocrine diseases (such as hyperthyroidism, diabetes, etc.); (5) patients with other immune system diseases (such as systemic lupus erythematosus, rheumatoid arthritis, Sjogren’s syndrome, etc.).

### 2.2. Determination of Vertebral Fractures

As some subjects had participated in our previous studies, information about their medical history, family history of osteoporotic fractures, history of fractures and recurrent falls, age at menopause, pregnancies and labors, diet, and physical activity was collected using a questionnaire on their first visit and through a review of hospital documents [12]. For the new participants recruited in this study, reports of incident fractures of the hip, vertebrae, or forearm after menopause were collected from the participants and subsequently confirmed by hospital documents, radiographic reports and other relevant sources. In addition, lateral X-rays of the spine covering the fourth thoracic spine to the fifth lumbar spine were performed for all new participants at the time of inclusion to detect any asymptomatic vertebral fractures. The fractures were reviewed by the same radiologist who did not take part in the genotyping or subsequent statistical analyses. A semiquantitative assessment of each vertebra from T4 to L4 according to Genant method was performed as follows: grade 0 (normal, no vertebral fracture); grade 1 (vertebral height reduction of 20% to 25%); grade 2 (between 25% to less than 40%); grade 3 (more than 40%) [13]. The fracture grades of all subjects were among grade 0 to grade 3. 

### 2.3. Basic Information and Clinical Measurements

All demographics and clinical information, including age, weight, body mass index (BMI), medical history, family history of osteoporotic fractures, history of fractures and recurrent falls, age at menopause, pregnancies and labors, diet, and physical activity were collected using both questionnaires during patients’ first visits and medical records review [12]. The serum level of calcium, phosphorus, alkaline phosphatase, 25-hydroxyl vitamin D, parathyroid hormone, liver and kidney function, procollagen type I N propeptide (PINP) and C-terminal cross-linking telopeptide of type I collagen (β-CTX) were also measured during the first visit. The serum level of periostin was measured using an enzyme-linked immunosorbent assay (ELISA) kit (human periostin/OSF-2 Duo Set ELISA; R&D Systems, Minneapolis, MN, USA). According to the product instructions, periostin prepared at 50 ng/mL was assayed and exhibited no cross-reactivity or interference. The measurements of circulating levels of periostin were taken at least 3 months post diagnosis. For the new participants, after they were recruited in our study, the measurements were taken on the first visit.

### 2.4. SNP Selection and Genotyping

In this study, 10 SNPs of *POSTN* were chosen from the Chinese Han (CHB) population in the 1000 Genomes Project (phase 3), with minor allele frequency (MAF) over 0.01 and linkage disequilibrium (LD) of SNPs higher than 0.8 (*r*^2^ > 0.8).

### 2.5. Measurement of BMD

BMD measurements (g/cm^2^) at the lumbar spine 1–4 (L1––4), the left femoral neck, the left trochanter and the total hip were detected by a dual-energy X-Ray absorptiometry (DXA, Lunar Prodigy Madison, WI, USA). The variation coefficients of BMD of lumbar spine 1–4, femoral neck and total hip were 1.40% [14], 2.23%, and 0.69% [15].

## 3. Statistical Analysis

Differences in clinical and BMD-related factors between patients with and without vertebral fractures were tested using unpaired t-test or Mann-Whitney U test as appropriate for continuous variables. All continuous variables were presented as mean ± SD or median (interquartile range). Allelic and genotypic logistic regressions adjusting for age, BMI, and BMD at the lumbar spine 1–4 were conducted using PLINK to identify associations between each SNP and the prevalence of vertebral fracture while accounting for multiple tests. ANOVA tests were also performed to find associations between polymorphisms in *POSTN* and serum periostin levels and BMD. Additionally, a quantile regression analysis was performed to find association between serum periostin levels and BMD using STATA.15.1. A multiple logistic regression analysis was used to find association between serum periostin levels, genetic polymorphisms in *POSTN* and the prevalence of vertebral fractures. All statistical tests were conducted using IBM SPSS Statistics 21.0 (IBM, Armonk, NY, USA) unless otherwise specified.

## 4. Results

### 4.1. Basic Characteristics of All Participants

The detailed information is presented in Table 1. The median age of all participants in this study was 65.74 years. Among 385 participants, 70 suffered at least one vertebral fracture (vertebral fracture subgroup), and 315 were without vertebral fractures (control subgroup). Compared with subjects without vertebral fractures, subjects with vertebral fractures were older (68.10 ± 9.38 vs. 65.74 ± 9.70, *p* = 0.024) and shorter (152.70 ± 5.92 vs. 154.30 ± 6.03, *p* = 0.018). Furthermore, the serum levels of periostin, alkaline phosphatase and phosphorus were significantly different between these two subgroups. In addition, BMD at the lumbar spine 1–4, the trochanter, the femoral neck, and the total hip were all significantly lower among patients with vertebral fractures compared to those without vertebral fractures (All *p* values < 0.001).

### 4.2. SNPs Distribution of POSTN and Minor Allele Frequencies (MAF)

In this study, 10 tag SNPs (MAF > 0.01) in *POSTN* were genotyped. All of them were in accordance with Mendelian genetics, and all genotype distributions followed the law of Hardy-Weinberg equilibrium (Table 2).

### 4.3. Association between Genetic Polymorphisms in POSTN and Serum Periostin Levels, BMD and the Prevalence of Vertebral Fracture

Among the 10 tag SNPs, the minor allele A carrier of rs9603226 had a significantly higher prevalence of vertebral fracture than wild type G carriers [OR (95% CI) = 1.722 (1.190–2.493), *p* = 0.037] by adjusting for age, BMI, and BMD at the lumbar 1–4. Furthermore, based on the genotypic association, we found that A/A homozygous of the same SNP had over three times the prevalence of vertebral fracture, compared to G/G homozygous [OR (95%CI) = 3.371 (1.318–8.617), *p* = 0.011] (Table 3). Moreover, we analyzed the associations between each SNP and serum periostin levels and BMD. We found that G/G homozygous of rs3923854 had higher serum periostin levels (67.26 ± 19.90) than C/C (54.57 ± 21.44, *p* = 0.036) and C/G (54.34 ± 18.23, *p* = 0.040) homozygous (Table 4). However, no association was found between any SNP in *POSTN* and BMD.

### 4.4. Association between Serum Periostin Levels and the Prevalence of Vertebral Fractures and BMD

Negative correlations between serum periostin levels and BMD at hip sites were found based on the median quantile linear regression. We found that for every thousand unit increase in serum periostin level, the predicted value of BMD at trochanter and total hip were decreased by 0.72 g/cm^2^ and 1.08 g/cm^2^ in the 75% regression, respectively. (Figure 1). The prevalence of vertebral fractures was significantly associated with BMD at lumbar spine 1–4 trochanter, femoral neck and total hip. However, there is no significant association between serum periostin levels and the prevalence of vertebral fractures.

## 5. Discussion

Periostin is a multifunctional soluble protein, which is widely expressed in human bones, periodontal ligaments, heart valves, and other fibrous connective tissues [16]. Its encoding gene, *POSTN,* was located in the chromosome 13q13.3 region, which was first discovered in 1999 by Horiuchi K. and colleagues in the preparation of cDNA libraries of MC3T3-E1 cells in mice [17]. The expression is most active in the process of osteogenesis or after mechanical stimulation. In bone metabolism, the transcription of *POSTN* is not only regulated by TWIST, RUNX2, and C-FOS/AP1, but also by PTH, growth factors (such as TGF-β, BMP2), cell growth factors (such as IL-4, IL-13), mechanical force, and a series of effects on the homeostasis of skeletal cells’ regulation [4]. Periostin plays at least two important roles in the field of bone metabolism. First, during the process of the formation of the original fiber, periostin enhances the activity of the Lysyl oxidase which is needed for collagen crosslinking [18]. The activity was achieved through the interaction of periostin with related proteins such as type I collagen, fibrin, Notch1, tendon protein-C and BMP-1. Second, periostin activates integrin-mediated signal transduction, thereby promoting cell adhesion and movement by activating the mechanism of actin/myosin contraction [19].

The first clinical trial on the association between serum periostin levels and the risk of osteoporotic fractures were performed by Rousseau and colleagues in 2014. They found an increasing risk of fracture in patients with high serum periostin levels [7]. In addition, Kim and colleagues concluded that serum periostin may be a potential biomarker for predicting the risk of osteoporotic fracture, especially for non-vertebral sites [8]. In 2012, Xiao SM et al. verified the relationship between the polymorphism of *POSTN* and BMD and confirmed the relationship between rs9547970 and vertebral fracture and BMD [11]. In 2017, Bonnet and colleagues verified the relationship between polymorphism of rs9547970 sites and BMD. They confirmed the interaction between the rs648438 of *LRP5* and the rs9547970 of *POSTN* in the porosity of serum periostin and radial cortex [10]. Moreover, they developed a new ELISA for a *K-POSTN*, through which they found that *K-POSTN* rather than total periostin was associated with fractures after adjusted by BMD, FRAX, bone microstructure or BTMs in 2017 [20]. J. Yan et.al and M. Farrokhiet.al found that higher serum periostin levels were presenting with acute hip fracture, implying periostin’s healing function during early phase [21]. However, the relationship among serum periostin levels, and the polymorphism of *POSTN*, BMD, and osteoporotic fractures at the same time in the same postmenopausal population need to be clarified. Therefore, our study focused on the relationship between the polymorphisms of *POSTN*, serum periostin levels, and the prevalence of osteoporotic vertebral fractures and BMD in Chinese postmenopausal women.

In our study, based on the quantile regression analysis, serum periostin levels were inversely associated with BMD at hip sites, which was comparable to the results based on the HKSC cohort with extreme BMD [7]. Serum periostin levels were nominally significantly associated with the prevalence of vertebral fracture. Although the significance was lost after being adjusted by BMD, serum periostin levels still could be a plausible surrogate marker of vertebral fractures prevalence, especially when BMD measurements at the spine cannot be measured because of the compression of vertebrae. We also found that rs9603226 in *POSTN* was associated with vertebral fractures, but no association was found between *POSTN* variation and BMD. There may be several reasons: First, the number of subjects included in the present study was only 385, therefore, the power of statistical was limited. Second, we found another SNP (rs9603226), which was not the same as rs9547970 identified by Xiao SM and Bonnet and their colleagues [11,20]. But both of them were in the same *POSTN* coding region: rs9603226 was located in chromosome 13.q13 38143586, and rs9547970 was located in chromosome 13.q13 38175191. Both of them were found the be related to alteration of transcription factor levels as well as methylation binding sites [22]. Different races and lifestyles could be possible reasons for results. Lastly, vertebral fractures and BMD are also associated with the risk of fall, environmental factors, nutritional status (especially calcium and vitamin D intake), and many complex factors such as exercise habits and sex.

Our study had some limitations. Apart from the fact that the number of participants recruited in our study was not a lot, additional factors such as smoking, exercise, nutritional factors, and other variables should be considered. Vertebral fracture grade should be classified. Moreover, the period of the vertebral fractures in our study was not the acute phase, that may account for no significances between two subgroups.

In conclusion, we found that the rs9603226 in *POSTN* was associated with the prevalence of vertebral fractures. Moreover, serum periostin levels were negatively associated with BMD at hip site. Therefore, serum periostin and its encoding gene could be informative clinical tools for predicting the risk of osteoporosis and vertebral fractures in Chinese postmenopausal women. Large sample-size clinical studies are needed to validate the results in the future. 

## Figures and Tables

**Figure 1 genes-13-00439-f001:**
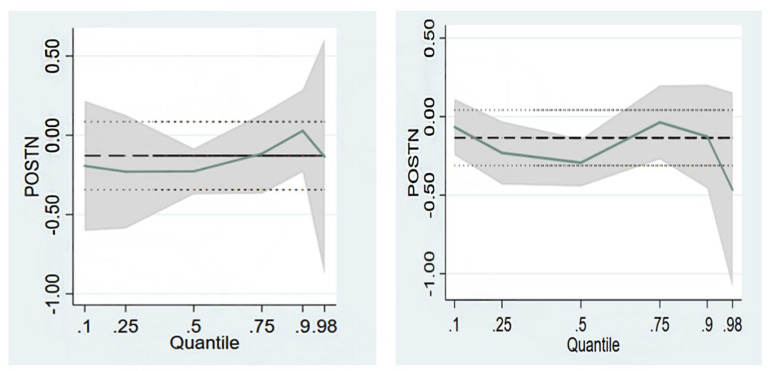
Association between serum periostin levels and BMD at hip sites (after adjusted by age). Figure 1 shows the association between serum periostin levels and BMD at different hip sites. It represented fitted conditional quantiles of BMD for a polynomial regression model in serum periostin levels. It showed different coefficients for periostin levels, corresponding to its modeled percentiles. The figure legend was changed and including additional explanation about the decrease in BMD was relatively small when periostin levels were low and became significant when high serum periostin levels were observed. The change in BMD was stable when periostin levels were low and significant BMD decrease was expected when higher serum periostin levels were observed. From left to right: serum periostin levels and BMD at trochanter, serum periostin levels and BMD at total hip.

**Table 1 genes-13-00439-t001:** Basic information, biochemical measurements and BMD of all subjects.

Variables	All Subjects (*n* = 385)	Vertebral Fracture Subgroup (*n* = 70)	Control Subgroup (*n* = 315)	*p* Value
Age (year)	65.74 ± 9.70	68.10 ± 9.38	65.22 ± 9.71	0.024 *
Height (cm)	154.30 ± 6.03	152.70 ± 5.92	154.65 ± 6.01	0.018 *
Body weight (kg)	56.33 ± 8.09	56.01 ± 8.56	56.40 ± 8.00	0.727
BMI (kg/m^2^)	23.65 ± 3.11	23.99 ± 3.28	23.58 ± 3.07	0.336
Serum calcium (mmol/L)	2.38 ± 0.48	2.44 ± 0.79	2.36 ± 0.38	0.257
Serum phosphorus (mmol/L)	1.17 ± 0.15	1.13 ± 0.15	1.18 ± 0.15	0.023 *
Adjusted ^#^	1.16 ± 0.20	1.18 ± 0.35	1.13 ± 0.18	0.037 *
Serum parathyroid hormone (mmol/L)	46.62 ± 19.24	46.46 ± 25.26	46.65 ± 17.68	0.953
Serum alkaline phosphatase (mmol/L)	76.75 ± 22.45	81.67 ± 26.96	75.63 ± 21.18	0.042 *
Adjusted ^#^	78.72 ± 28.85	81.85 ± 52.19	75.59 ± 25.10	0.036 *
Serum creatine (mmol/L)	58.13 ± 11.41	57.63 ± 11.96	58.27 ± 11.29	0.730
Serum urea (mmol/L)	5.27 ± 1.56	5.16 ± 1.24	5.30 ± 1.64	0.562
Serum 25-hydroxylvitaminD (mmol/L)	11.64 ± 6.95	12.13 ± 7.30	11.53 ± 6.88	0.515
Serum periostin (ng/mL)	55.09 ± 21.03	59.57 ± 26.05	54.10 ± 19.65	0.049 *
Adjusted ^#^	56.64 ± 27.14	59.07 ± 49.21	54.21 ± 23.09	0.081
Serum PINP (ng/mL)	57.95 (40.06–70.66)	57.42 (46.76–78.76)	54.56 (38.96–69.65)	0.162 ^1^
Serum β-CTX (ng/mL)	464.80 ± 225.97	430.09 ± 213.94	472.52 ± 228.16	0.156
BMD at lumbar spine 1–4 (g/cm^2^)	0.86(0.79–0.94)	0.78(0.63–0.88)	0.87(0.80–0.94)	<0.001 ** ^1^
Sqrt L1-L4 BMD Adjusted ^#^	0.92 ± 0.11	0.90 ± 0.20	0.94 ± 0.07	0.001 *
BMD at trochanter (g/cm^2^)	0.59 ± 0.10	0.53 ± 0.12	0.60 ± 0.09	<0.001 **
Adjusted ^#^	0.57 ± 0.11	0.54 ± 0.23	0.60 ± 0.11	<0.001 **
BMD at femoral neck (g/cm^2^)	0.71 ± 0.11	0.65 ± 0.13	0.72 ± 0.10	<0.001 **
Adjusted ^#^	0.69 ± 0.13	0.67 ± 0.25	0.72 ± 0.11	<0.001 **
BMD at total hip (g/cm^2^)	0.75 ± 0.12	0.69 ± 0.15	0.77 ± 0.10	<0.001 **
Adjusted ^#^	0.73 ± 0.13	0.70 ± 0.25	0.77 ± 0.11	<0.001 **

Sqrt, square root calculations; PINP, procollagen type I N propeptide; β-CTX, β isomerized C-terminal crosslinking to type I collagen; BMD, bone mineral density. ^#^ Adjusted by age. * *p* < 0.05 was considered as statistically significant. ** *p* < 0.001 was considered as statistically significant. ^1^ Using Mann-Whitney U tests.

**Table 2 genes-13-00439-t002:** Detailed information of ten tag SNPs of *POSTN*.

SNP	Alleles	Minor Allele	MAF in This Study	MAF in CHB ^^^
rs9547952	C/T	T	0.088	0.068
rs9603226	A/G	A	0.371	0.422
rs7322993	C/T	T	0.139	0.126
rs9547965	A/G	A	0.045	0.068
rs1924285	A/T	A	0.139	0.126
rs7338244	C/G	G	0.273	0.262
rs3923854	C/G	G	0.134	0.136
rs2025405	C/T	T	0.273	0.262
rs3829365	G/C	C	0.355	0.316
rs9547970	A/G	G	0.226	0.194

^^^ CHB: Chinese Han in Beijing.

**Table 3 genes-13-00439-t003:** Association between ten tag SNPs of *POSTN* and the risk of vertebral fractures (*n* = 385).

SNP	Allelic OR (95% CI)	Genotypic OR (95% CI)	*n*
rs1924285	0.642 (0.354–1.163)	T/T 1	288
		A/T 0.795 (0.418–1.510)	87
		A/A NA	10
rs2025405	0.827 ( 0.542–1.264)	C/C 1	212
		C/T 0.894 (0.512–1.560)	136
		T/T 0.652 (0.239–1.775)	37
rs3829365	0.654 (0.437–0.978)	G/G 1	158
		G/C 0.647 (0.375–1.114)	181
		C/C 0.413 (0.152–1.124)	46
rs3923854	1.098 (0.649–1.860)	C/C 1	295
		C/G 1.460 (0.795–2.681)	77
		G/G 0.399 (0.051–3.135)	13
rs7322993	0.642 (0.354–1.163)	C/C 1	288
		C/T 0.795 (0.418–1.510)	87
		T/T NA	10
rs7338244	0.827 (0.542–1.264)	C/C 1	212
		C/G 0.894 (0.512–1.560)	136
		G/G 0.652 (0.239–1.775)	37
rs9547952	0.961 (0.501–1.845)	C/C 1	324
		C/T 1.021 (0.486–2.145)	54
		T/T 0.749 (0.088–6.335)	7
rs9547965	1.354 (0.602–3.046)	G/G 1	351
		G/A 1.492(0.643–3.462)	33
		A/A NA	1
rs9547970	0.743 (0.467–1.181)	A/A 1	237
		G/A 0.681 (0.377–1.232)	122
		G/G 0.716 (0.236–2.175)	26
rs9603226	1.722 (1.190–2.493) *p* = 0.037 *^,&^	G/G 1	151
		G/A 1.617 (0.732–3.575)	182
		A/A 3.371 (1.318–8.617) *^,#^ *p* = 0.011	52

* *p* < 0.05 was considered as statistically significant. ^&^ Adjusted *p* value (Bonferroni correction). ^#^ Adjusted by age, BMI and BMD at lumbar 1–4.

**Table 4 genes-13-00439-t004:** Association between genetic polymorphisms of *POSTN* and serum periostin level and BMD (*n* = 385).

SNP (Genotype)	*n*	Serum Periostin Level (ng/mL)	BMD at Lumbar Spine 1–4 (g/cm^2^)	BMD at Femoral Neck (g/cm^2^)	BMD at Trochanter (g/cm^2^)	BMD at Total Hip (g/cm^2^)
rs1924285						
T/T	288	55.06 ± 20.64	0.85 ± 0.20	0.71 ± 0.11	0.59 ± 0.11	0.75 ± 0.12
A/T	87	55.19 ± 22.65	0.84 ± 0.22	0.71 ± 0.11	0.59 ± 0.09	0.76 ± 0.10
A/A	10	55.09 ± 19.43	0.86 ± 0.15	0.72 ± 0.15	0.62 ± 0.10	0.78 ± 0.15
rs2025405						
C/C	212	54.81 ± 21.09	0.84 ± 0.20	0.71 ± 0.11	0.58 ± 0.11	0.75 ± 0.12
C/T	136	54.02 ± 21.44	0.85 ± 0.20	0.71 ± 0.11	0.59 ± 0.09	0.76 ± 0.10
T/T	37	60.66 ± 18.67	0.87 ± 0.21	0.71 ± 0.12	0.59 ± 0.11	0.76 ± 0.13
rs3829365						
G/G	158	54.47 ± 20.82	0.84 ± 0.21	0.71 ± 0.11	0.59 ± 0.10	0.75 ± 0.12
C/G	181	53.58 ± 21.89	0.84 ± 0.20	0.71 ± 0.11	0.59 ± 0.11	0.76 ± 0.12
C/C	46	52.89 ± 17.58	0.85 ± 0.20	0.71 ± 0.09	0.59 ± 0.10	0.75 ± 0.11
rs3923854						
C/C	294	54.57 ± 21.44	0.84 ± 0.20	0.71 ± 0.11	0.59 ± 0.10	0.75 ± 0.12
C/G	77	54.34 ± 18.23	0.85 ± 0.23	0.71 ± 0.11	0.59 ± 0.09	0.75 ± 0.11
G/G	13	67.26 ± 19.90 *^,&,#^	0.92 ± 0.09	0.74 ± 0.11	0.60 ± 0.11	0.77 ± 0.12
rs7322993						
C/C	288	55.06 ± 20.64	0.85 ± 0.20	0.71 ± 0.11	0.59 ± 0.11	0.75 ± 0.12
C/T	87	55.19 ± 22.65	0.84 ± 0.22	0.71 ± 0.11	0.59 ± 0.09	0.76 ± 0.10
T/T	10	55.09 ± 19.43	0.86 ± 0.15	0.72 ± 0.15	0.61 ± 0.10	0.78 ± 0.15
rs7338244						
C/C	212	54.81 ± 21.09	0.84 ± 0.20	0.71 ± 0.11	0.58 ± 0.11	0.75 ± 0.12
C/G	136	54.02 ± 21.44	0.85 ± 0.20	0.71 ± 0.11	0.59 ± 0.09	0.76 ± 0.10
G/G	37	60.66 ± 18.67	0.87 ± 0.21	0.71 ± 0.12	0.59 ± 0.11	0.76 ± 0.13
rs9547952						
C/C	324	54.83 ± 21.30	0.84 ± 0.21	0.71 ± 0.11	0.59 ± 0.10	0.75 ± 0.12
C/T	54	55.75 ± 19.81	0.87 ± 0.16	0.70 ± 0.11	0.58 ± 0.09	0.75 ± 0.11
T/T	7	62.15 ± 18.18	0.90 ± 0.10	0.78 ± 0.07	0.65 ± 0.10	0.82 ± 0.08
rs9547965						
G/G	351	54.78 ± 21.16	0.84 ± 0.19	0.71 ± 0.11	0.59 ± 0.10	0.76 ± 0.12
G/A	33	58.08 ± 19.81	0.84 ± 0.30	0.71 ± 0.11	0.58 ± 0.10	0.74 ± 0.12
A/A	1	67.87	0.96	0.75	0.64	0.83
rs9547970						
A/A	237	54.43 ± 20.72	0.84 ± 0.21	0.71 ± 0.11	0.58 ± 0.11	0.75 ± 0.12
G/A	122	56.19 ± 22.55	0.86 ± 0.21	0.71 ± 0.11	0.59 ± 0.09	0.76 ± 0.11
G/G	26	55.95 ± 16.36	0.86 ± 0.12	0.71 ± 0.12	0.60 ± 0.11	0.76 ± 0.13
rs9603226						
G/G	151	54.75 ± 20.40	0.85 ± 0.20	0.71 ± 0.10	0.59 ± 0.09	0.76 ± 0.11
G/A	182	54.60 ± 21.03	0.85 ± 0.20	0.71 ± 0.12	0.59 ± 0.11	0.76 ± 0.12
A/A	52	57.83 ± 22.93	0.81 ± 0.22	0.70 ± 0.11	0.58 ± 0.10	0.74 ± 0.12

* *p* < 0.05 was considered as statistically significant. ^&^ C/C versus G/G. ^#^ C/G versus G/G.

## Data Availability

The data presented in this study are available on request from the corresponding author. The data are not publicly available due to privacy.

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
