# Peer review of "Serum Periostin Level and Genetic Polymorphisms Are Associated with Vertebral Fracture in Chinese Postmenopausal Women"

_genes, 2022, doi:10.3390/genes13030439_

Round 1

Reviewer 1 Report

The article "Serum periostin level and genetic polymorphism are associated with a vertebral fracture in Chinese postmenopausal women" investigates the association between periostin level/gene polymorphism and vertebral fracture in the Chinese population. Unfortunately population sample was quite small and the control group was 4.5 times bigger than the subgroup with vertebral fractures. As the mean age of the subgroup with vertebral fracture was higher it cannot be excluded the risk of fracture was mainly attributed to the age of the investigated females. 

I advise to submit the article as the preliminary results short report, as it correlation between polymorphism dn periostin level requires further investigation and a bigger population sample.

Reviewer 2 Report

The authors have performed an association study to investigate the relations between serum periostin levels, some variations of its encoding gene (POSTN), bone mineral density (BMD) and the prevalence of vertebral fractures. The objective is clear, but I have some comments about the methodology and results presented:

1 - In the study author have included 385 participants, 70 with at least one vertebral fracture, and 315 without vertebral fractures (control subgroup). Among the control subgroup, Are they diagnosed by osteoporosis? Or not-osteoporotic postmenopausal women? Because their BMD is significantly higher in all regions and they have not fractures by fragility. Have they other fractures? hip fractures maybe?
Explain this please, because if not, you are comparing osteoporotic with vertebral fractures Vs. non-osteoporotic postmenopausal women. So the final conclusion is relevant because the associations found could be related with the osteoporotic status, not the fractures.

2 - In line with the comment above. Authors have found significant serum periostin differences, which lost the significance when adjusted by BMD. Explain this, because it is important for the conclusion: "serum periostin and its encoding gene could be informative clinical tools for predicting the risk of osteoporosis and vertebral fractures in Chinese postmenopausal women".

3 - In the section 4.4 (Line 174). Authors declare that negative correlations between serum periostin levels and BMD at hip sites were found based on the median quantile linear regression.

Please, give the results which support this statement. With the tables above I cannot see it clearly. The median quantile regression plot will suits perfectly in this section.

Other minor corrections:

-The introduction has a poor background. Please improve it.

-Line 49. POSTN polymorphisms *are* associated ...

-Along the whole text 'lumber spine', please correct it -> 'lumbar spine'

-Line 194. periostin enhances the activity of the *ampicillin* oxidase which is... Do you mean Lysyl oxidase? Please revise it.

-Line 206. Revise the citation, because it is wrong. It does not correspond to Xiao SM et al. study.
